# Effect of the First Feeding on Enterocytes of Newborn Rats

**DOI:** 10.3390/ijms232214179

**Published:** 2022-11-16

**Authors:** Maria A. Nikonova, Irina S. Sesorova, Ivan D. Dimov, Natalia R. Karelina, Alexander A. Mironov

**Affiliations:** 1Department of Anatomy, Ivanovo State Medical Academy, 153012 Ivanovo, Russia; 2Department of Anatomy, Saint Petersburg State Pediatric Medical University, 194100 Saint Petersburg, Russia; 3The AIRC Institute of Molecular Oncology, 20139 Milan, Italy

**Keywords:** Golgi complex, enterocytes, newborn, lipid overloading, lymphatic capillaries, transcytosis

## Abstract

The transcytosis of lipids through enterocytes occurs through the delivery of lipid micelles to the microvilli of enterocytes, consumption of lipid derivates by the apical plasma membrane (PM) and then their delivery to the membrane of the smooth ER attached to the basolateral PM. The SER forms immature chylomicrons (iChMs) in the ER lumen. iChMs are delivered at the Golgi complex (GC) where they are subjected to additional glycosylation resulting in maturation of iChMs. ChMs are secreted into the intercellular space and delivered into the lumen of lymphatic capillaries (LCs). The overloading of enterocytes with lipids induces the formation of lipid droplets inside the lipid bilayer of the ER membranes and transcytosis becomes slower. Here, we examined components of the enterocyte-to-lymphatic barriers in newly born rats before the first feeding and after it. In contrast to adult animals, enterocytes of newborns rats exhibited apical endocytosis and a well-developed subapical endosomal tubular network. These enterocytes uptake membranes from amniotic fluid. Then these membranes are transported across the polarized GC and secreted into the intercellular space. The enterocytes did not contain COPII-coated buds on the granular ER. The endothelium of blood capillaries situated near the enterocytes contained only a few fenestrae. The LCs were similar to those in adult animals. The first feeding induced specific alterations of enterocytes, which were similar to those observed after the lipid overloading of enterocytes in adult rats. Enlarged chylomicrons were stopped at the level of the LAMP2 and Neu1 positive post-Golgi structures, secreted, fused, delivered to the interstitial space, captured by the LCs and transported to the lymph node, inducing the movement of macrophages from lymphatic follicles into its sinuses. The macrophages captured the ChMs, preventing their delivery into the blood.

## 1. Introduction

Our first paper supported the following model of lipid transcytosis. Briefly: Ingested triglycerides are hydrolyzed by gastric and pancreatic lipases and mixed with bile acids obtained from the liver to form micelles consisting of free fatty acids and monoglycerides and triacyl glycerides (TAG) [1,2]. During their hydrolysis, fatty acids and monoglycerides are formed in the intestinal lumen, which form mixed micelles with bile salts. Micelles move to the microvilli of enterocytes and come into contact with the APM, giving their components to the latter. Then, free fatty acids diffuse along the APM and flip-flop from the outer monolayer of the apical plasma membrane (APM) to the cytosolic one and then diffuse to the domains of the basolateral PM (BLPM), where they contact the tubular network of the smooth endoplasmic reticulum (SER). In this place, there are proteins that transfer lipids through the aqueous phase of the cytosol and deliver lipids to the membrane of the SER. These tubules form networks connected to the granular ER [3,4]. Delivery of free fatty acids, monoacylglycerides and cholesterol from the BLPM to these membranes ER occurs through lipid-carrying proteins [4,5]. 

In the SER, components of fat are reformed into triglycerides by fatty-acyl-CoA synthetase and various acyltransferases [2,6,7,8]. The cholesterol uptake by enterocytes depends on the C1-like Niemann–Pick protein 1 [9,10] and the cholesterol transporter ABCA1 [11]. Cholesterol esters, phospholipids and ApoB48 form immature chylomicrons (iChMs) in the lumen of the endoplasmic reticulum (ER) [12]. In the small intestine enterocytes, there are no COPII-coated buds and typical ER exit sites on the granular ER [3,4]. 

Next, iChMs are transported to the Golgi complex (GC), where ApoB48 is subjected to glycosylation [10,13]. During intra-Golgi transport, iChMs undergo glycosylation and are concentrated on the trans side of the Golgi stack. Their secretion occurs mainly into the intercellular space in the area of interdigital contacts (IDCs), where actomyosin cuffs surround IDCs. These cuffs can compress the intercellular space inside the contacts to force the mature ChMs (mChMs) to move towards the BM [3,4].

Apical endocytosis is poorly developed in the enterocytes of adult animals [3,4]. A unique feature of enterocytes in newborn mammals is the presence of the apical tubular endosomal system [14]. Membranes captured from amniotic fluid are absorbed by enterocytes apically, transported to the endoplasmic reticulum and converted into fatty acid triglycerides-CoA synthetase and various acyltransferases [2,8]. In young mammals, maternal IgGs pass through the glycocalyx, bind to the receptors of the Fs fragments of antibodies that are located on the apical plasma membrane, and then pull up to the base or diffuse through the glycocalyx directly to the bases of the microvilli, where the concentration of Fs fragments receptors is higher [14,15,16,17,18,19,20]. Then they are endocytosed via clathrin-coated pits or via noncoated pits and invaginations and transported to the intercellular space.

In adult animals, the delivery of mChMs to the interstitial space of an intestinal villus occurs through the pores in the basement membrane (BM), which are formed by the processes of dendritic cells. In adult animals, the network of blood capillaries is situated in close contact with the BM of enterocytes. This part of the blood capillary wall that faces the enterocytes contains a large number of fenestrae [4]. These blood capillaries cannot absorb chylomicrons due to size exclusion [21,22]. The relative length of the lymph capillaries (LCs) is fixed at ~70% of the length of the villi [22,23,24]. LCs exhibited close contacts with smooth muscle cells (SMCs) [22,23,24,25,26].

Then, mChMs are absorbed into the lumen of lymph capillaries through their specific intercellular contacts [4]. The passage of ChMs depends on the contractility of smooth muscle cells, which form longitudinal and spiral bundles in their own plate. Additionally, the contractility depends on intestinal wall peristalsis and longitudinal and/or lateral contractions of villi [25,27,28,29,30,31]. The subsequent SMC relaxation is necessary for the active absorption of ChMs [30]. This contractility also ensures the capture of ChMs from the interstitial space into the lumen of lymphatic capillaries (LCs) through contacts between ECs [4]. 

Mesenteric collecting vessels, like all lymphatic collecting vessels, are constantly pumping via coordinated SMC contractions through lymphatic valves, except in mice, in which mesenteric lymphatic vessel pumping is not observed [22,32,33,34,35,36]. Increased fat uptake induces augmentation of the size of iChMs, but not to an increase in the number of chylomicrons [3,13].

LCs are important for the survival of, at least, mice. Indeed, the loss of intestinal lymph nodes led to severe intestinal inflammation, sepsis and high mortality [37]. Of interest, mice with diet-induced obesity have larger drops of lipids in the intestinal cytoplasm compared to lean mice [38,39]. All this proves the outstanding, but still underestimated role of intestinal LC functioning in close connection with enterocytes [22].

The study of the mechanisms of lipid consumption in the small intestine is very important because a large number of monogenic hereditary diseases associated with mutations of proteins involved in lipid absorption have been identified. The widespread occurrence of atherosclerosis and obesity among the population of economically developed countries raises the question of the mechanisms of fat absorption among the priorities. A new hypothesis [5,6] explaining the development of atherosclerosis by excessive fat eating per once is based on observations that when enterocytes are overloaded with lipids, glycosylation errors of ChMs appear after the excessive eating of lipids at a time [3,4,40,41,42,43,44,45,46].

Here, we examined enterocytes in newborn rats and transport routes in the enterocyte responsible for lipid absorption.

## 2. Results

Our study was initiated by the presence of a contradiction between the detection of poorly developed apical endocytosis in the enterocytes of adult rats with literature data showing that apical endocytosis is well expressed in young animals. It was necessary to check whether there is pronounced endocytosis in newborn rats that have not yet received their first milk from their mother, at the so-called zero point.

At the same time, it was necessary to understand to what extent the structures involved in the capture of lipids transferred through enterocytes are differentiated. To do this, we studied the structure of the microcirculatory bed of the intestinal villi in the newborn rats.

In adult animals, the intestinal villi in the jejunum have the form of narrow folds (leaf-like outgrowths). The folds are elongated perpendicular to the long axis of the small intestine [3]. The shape factor increases from the tip to the base (Table 1). Light and dark enterocytes are clearly visible (Appendix A: indicated by black arrows). The microvilli of the enterocytes of newborns are shorter than the microvilli of the enterocytes of adult rats and have cylindrical shape (Figure 1A,F,J). Enterocytes of adult rats have a prismatic shape with a height of 25 µm (Figure 1I).

The structure of the vascular bed of the intestinal villi of newborn rats before their feeding was less developed than in adult animals (Figure 1B–E). This was visible in corrosion casts of blood microvessels in intestinal villus of newborn rats and in total preparations of intestinal villi after their staining for horseradish peroxidase (blood capillaries contain black precipitate of DAB-HRP reaction in their lumen; Figure 1G,H). The number of blood capillaries on a cross-section of the villus of the jejunum increased from 2 near the apex to 8–10 near the villus bottom (Table 2). The number of lymphatic capillaries (LCs) in the intestinal villi of an adult rat varies from one to four. There are anastomoses between them (Table 3) [4,30]. LCs are located closer to the central part of the villi and have the appearance of a slit-like space limited by endothelial cells that do not have a BM (Table 3). Appendix A demonstrate for comparison serial images of the longitudinal section of enterocytes obtained with 3VIEW. Also, Appendix A show the rotation of the 3D model of the tubular networks formed by the smooth ER near the BLPM (brown planes; yellow structures are endosomes; the ER is colored in green; light brownish vacuoles represent multivesicular body). The ER forms contacts sites with the BLPM. Appendix A demonstrate corrosion casts and total preparations of the blood microvessels of the villi (compare with Figure 1B–E). Thus, we found a weak development of blood capillaries and the absence of fenestra fields in them.

In newborn rats, the intestinal villi of the initial jejunum have the shape of an elongated cylinder which is slightly flattened. Their shape factor varied from 1.6 to 2 (Table 1). 

Figure 1A,F show scanning electron microscopy of the luminal surface of intestinal villi. In the newborn rats before their first feeding, enterocytes are slightly lower than the enterocytes of adult rats (Figure 1I). The percentage of enterocytes with pronounced endocytosis and subapical networks is higher in newborn rats (Figure 1M,N). Small membrane residues in the form of fragments or bubbles pass between the microvilli and are captured in the clathrin-coated buds. Similar membranes were also observed between enterocytes. Large undifferentiated interdigital contact (arrow) between two enterocytes was observed (Figure 1O). Tight junctions were normal. Newborn rats had a significantly lower number of contact sites between the smooth ER and the BLPM (Figure 1K, Figure 2A,B and Figure 3F).

Thus, we have shown for the first time that in newborns, apical endocytosis is well expressed and is used for transcytosis of membranes absorbed from amniotic fluid. These membranes are delivered to the endosomes, then appeared in the Golgi cisternal distensions and are secreted into the space between the enterocytes.

In newborn rats, we did not find a well-developed smooth ER attached to the BLPM in the space between the tight junctions and the level of the location of the GC described earlier [3,4]. This questions the ability of neonatal enterocytes to transport lipids from gut to interstitial space. The absence of the SER attached to the BLPM spoke against the fact that the movement of lipids is carried out through the membranes. However, these enterocytes could uptake lipids (see below). Therefore, we studied the structure of these enterocytes after the first feeding and found that lipids are absorbed by these cells, but transcytosis of lipids is significantly delayed at the level of GC and post-GC. Consequently, there was another way for lipids to move from the APM into the ER membranes, or they could diffuse through the aqueous phase. Therefore, we performed a three-dimensional reconstruction of the apical endocytic tubular network. 

Serial tomographic images in Figure 2A,B demonstrate the subapical zone of young enterocytes. Figure 2C–F present the three-dimensional model obtained on the basis of EM tomography. View from different angles. ER is colored green. Rough ER is large vacuoles and cisterns. Yellow structures indicate endosomes. The purple color indicates APM. The BLPM is indicated in red. Multivesicular corpuscles and late endosomes are indicated in blue. Endosome–ER contact sites are shown with white arrows. Clathrin appears as purple triangles. 

Thus, we found well-developed contact sites between the endosome membrane and the ER membrane. The detection of contact sites between the smooth ER and endosomes in the apical part of enterocytes supported the previous hypothesis.

Figure 2F shows the enlarged area of the image enclosed inside the red box in Figure 2D. The Golgi stacks were already polarized. No COPII-coated buds were visible on the ER. Additionally, typical ERES’ were absent. Immune EM marking for GM130 was present on the cis side of GC. (Figure 2G,F). The labeling for sialyltransferase was observed at the trans side of the stacks (Figure 2H). Labeling for LAMP2 and GM130 in (Figure 2G) and LAMP-2 (10 nm) and Golgin-97 (15 nm) in (Figure 2H) were observed out of the Golgi stacks. In newborn rats, Golgi stacks were smaller than in adult animals (Figure 3G). Multilamellar structures are visible in the cisternal distension of the Golgi complexes. (Figure 3A–D). Membranes remnants were visible between enterocytes (Figure 3E), but also in the interstitial space (Figure 4B–D).

The walls of the blood capillary contained a low number of fenestrae. (Figure 1P and Figure 4A–D). The density of microvessels in an intestinal villus of newborn rats were lower than in adult rats (Figure 1B–E,G,H; Table 2). The mean number of adjacent enterocytes was lower and the distribution of them is shifted towards four (Figure 3H,I).

In newborn rats, LCs were wider than in adult animals (Table 3). The structure of inter-endothelial contacts (indicated by white arrows) in the LCs of a newborn before feeding did not significantly differ from that of adult animals (Figure 4E–G); white arrow indicates nervous terminal). Newborns have only one lymphatic capillary. It reaches only the middle third of the villi (our unpublished observations). The subcapsular sinuses of lymph nodes were empty (Figure 1P,Q,S).

By 15 min after the first feeding, the formation of large lipid droplets (white arrows) in the cytoplasm was observed (Figure 5A–C and Figure 6A–C). Huge lipid droplets and “lipid lakes” in enterocytes and between enterocytes were visible (Figure 5D and Figure 6D,F). Their peripheral part is more electron-dense. The percentage of such cells was equal to 25%. The GCs were overloaded (Figure 5E). Large ChM were observed in the post-Golgi compartment positive for Nau1 (Figure 5F).

In an attempt to understand where the delay of chylomicrons occurs in the enterocytes of newborns after the first feeding, we performed immunolabeling on LAMP2 and found that in post-GC carriers, there is labeling on LAMP2. Labeling for LAMP2, GM130 and ApoB of post-Golgi carriers containing large mChMs was also detectable (Figure 6G,H). 

The lumen of the blood capillary with the erythrocytes inside its lumen does not contain chylomicrons (Figure 6E).

Mechanisms of absorption of large chylomicrons from the interstitial space into the lymphatic capillaries of a newborn rat after feeding are rather similar with those observed in adult rats [3,4]. Large ChMs were observed in the inter-endothelial contacts (Figure 7F) and in the lumen of the lymphatic capillary. However, in contrast to adult animals, large ChMs were detectable in the lumen of the blood capillary (Figure 7H: fenestrae are visible in the wall), although rarely. Lipid droplets are visible in the interstitial space. The lumen of LCs was filled with larger ChMs (Figure 8A,B, white arrow). Interdigital contacts between endothelial cells in LCs did not allow larger ChMs (arrows) to pass through it (Figure 8B). The migration of macrophages into the subcapsular sinus of the lymph node and uptake of large chylomicrons by lymph node macrophages (Figure 8C–H).

Thus, the changes observed in the structures of the intestinal villi after the first feeding of newborn rats correspond to the phenotype of adult enterocytes when they are overloaded with lipids [3]. Hypothetical schemes explaining the mechanisms of lipid transcytosis in adult and newborn rats are shown in Figure 9 and Figure 10.

## 3. Discussion

Here, we studied the gut-to-lymph barrier immediately after birth (however, before the first feeding) and after the first feeding. Newborns have apical clathrin-dependent endocytosis, and a subapical endosomal plexus, whose tubules have contacts with tubes and cisterns of smooth ER, but there is no plexus of smooth ER tubules, some of which have contacts with the BLPM. There are no well-developed interdigitating contacts between enterocytes surrounded by the actin–myosin cuff. Already, the first feeding causes an overload of hypertension and (judging by the reaction with lectins) a violation of the glycosylation process. Lipid droplets appear in the cytoplasm of enterocytes, which indicates their overload. In addition, the blood pressure is overloaded with chylomicrons and large chylomicrons and even “lipid lakes” are formed between the enterocytes. In the interstitial, the chylomicrons are very large. They are absorbed into the lymphatic capillaries through the contacts of the sinusoid type between ECs [4,30,47,48,49], then, pass through the system of lymphatic vessels, and enter the blood. The ER is the place where nascent iChMs are formed with the inclusion of chylomicron-specific proteins, such as apolipoprotein B48 (intestinal-specific isoform encoded by ApoB) [12]. iChMs are transported to the GC for further processing, and mature chylomicrons are secreted into the space between the BLPM of enterocytes [3]. Similarly with adult rats, in newborn rats, enterocytes did not contain COPII-coated buds both before feeding and after it. This argues once more that a COPII-dependent vesicle does not exist in organisms, at least in enterocytes [50,51,52,53,54,55,56]. We observed features of the transporting GC, namely, in addition to the presence of osmiophilic particles in the extensions of the lateral edge of the cisterns, there is a tubulation of membranes at the cis and trans poles of the organelle; tubular connections between adjacent stacks and membrane carriers with lipids at the trans pole are revealed [52].

We demonstrated that in rats that received lipids in a limited amount after fasting, all systems involved in lipid transcytosis work normally. In contrast, after lipid overloading, a lot of lipid droplets were observed in enterocytes. When enterocytes are overloaded with lipids, transcytosis slows down, lipid droplets form in the cytoplasm of cells and the cisternae of Golgi complex are overloaded with ChMs. Lipid droplets are formed from tubular network of SER [57]. There were identified 181 proteins associated with the cytoplasmic lipid droplets fraction from rat enterocytes [38,39]. Impairment of lipid metabolism affects consumption of dietary fat [58].

It is known that under the conditions of high acidity of the medium, the heads of fatty acids lose their charge due to the addition of a proton and can jump from one monolayer of lipids to another. Taking into consideration our data showing that lipids are not observed in the cytosol [3] and considering that it is extremely difficult for cholesterol to cross the lipid bilayer from one water phase to another, we proposed that lipids enter the apical plasmalemma, diffuse along it, jump from the external leaflet to the cytosolic one and then transport from the BLPM at the ER membrane with the help of lipid transporters. Transcytosis of lipids through the epithelial layer of the jejunum is characterized by the absence of lipid uptake by endocytosis, the appearance of the first iChMs in the SER network adjacent to the BLPM below the subapical zone of the cell and their passage through the GC. 

It was proposed that in adults, the main mechanism responsible for the absorption of lipids is their incorporation into the apical membrane from micelles formed by bile acids, then their diffusion across the membrane and, finally, transport into the membranes of the ER with the help of carrier proteins [3]. In the current work, we found that newborn rats do not have a developed network of smooth ER near the BLPM, but there is apical endocytosis. It was logical to assume that these enterocytes would not be able to absorb lipids. However, it turned out that they absorb them, although they have a delay in removing lipids from the ER. They have a large number of lipid drops. Thus, after the first feeding, the phenotype of changes in enterocytes, microvascular bed and interstitial significantly differs from that described in adult animals when using low doses of lipids [3,4]. 

However, the tubular network of the smooth ER typical for adult rats and situated near the basolateral plasmalemma forming contact sites with the BLPM was purely developed in the newborn rats. In contrast, the purely developed endocytic structures in adult rats were well visible in newborn animals. If we assume that the consumption of lipids occurs through the contacts sites between the SER and BLPM, the obvious prediction would be to expect that these enterocytes would not be able to absorb lipids. Meanwhile, after the first feeding, we observed a large number of lipid droplets, which could be formed only if the lipids appear within the lipid bilayers of the ER membrane.

In order to understand what such dramatic changes in enterocytes and interstitial space are associated with, we compared these results with those obtained in adult rats that were overloaded with lipids. Many of these changes were also found there, but giant lipid droplets in the cytoplasm, chylomicrons draining between enterocytes and “lipid lakes” in the interstitial were not found. Among enterocytes, along with more or less intact cells, there were cells whose cytoplasm was sharply overloaded with large lipid droplets. 

Meanwhile, newborn rats had a lower number of contact sites between the SER with the BLPM. This raised doubts regarding our hypothesis suggesting that molecules of lipids enter enterocytes not through the cytosol but through the APM with subsequent diffusion and flip-flop along the lipid bilayer of the APM and BLPM. To test it, we performed a three-dimensional reconstruction of the subapical membranous tubular plexus and found contact sites between endosomes and cisterns of smooth ER which could compensate the lack of the ER–BLPM contact sites. Based on this observation, we assumed that, instead of contact zones with the plasmalemma, neonatal enterocytes use contact zones with endosomes to transfer lipids through the aqueous phase, which, as shown by three-dimensional reconstruction, are in close contact with the apical plasmalemma and it is always possible to establish membrane continuity. We calculated the average number of such contact zones in the enterocytes of adult and newborn hungry animals inside a 200 nm slice used for direct observation. We calculated this parameter directly under the microscope screen, including here both the contact zones between the smooth ER and BLPM and the same zones between smooth ER and endosomes. In newborns, the number of such contact sites was lower. Nevertheless, these contact sites explain why, with almost complete absence of contact zones between the BLPM and smooth ER, lipids enter the enterocyte and form lipid droplets.

The tubular–vesicular network of newborns is part of the endocytic cell pathway, which is connected to the secretory pathway of the cell. The tubular–vesicular network of newborns is well expressed and is present mostly within the subapical part of the cell. Membrane fragments found in the endosomes of the enterocyte and in the GC and between enterocytes could contain maternal proteins inducing tolerance in the young rats. Our data could explain why kidneys taken for transplantation from a mother who once fed this child are better adapted to these children [56].

Why do newborns need apical endocytosis? In order to receive antibodies from the mother. Usually these are G immunoglobulins. There is a Fc fragment at the base of the antibody, and it adheres to its receptor, which is expressed on the microvilli [14]. Then, the glued antibody is pulled up to the base of the microvilli and undergoes apical endocytosis. On the other hand, we must take into consideration that the tubular network composed of the SER is purely developed in newborns.

Usually, mChMs are secreted into the intercellular spaces between enterocytes in the areas of existing lock contacts. To get into the intestinal space (lamina propria), ChMs have to pass through the BM. It is known that the pores in the basement membrane are formed by dendritic and epithelial cells [3,4]. Dendritic cells with their cytoplasmic processes “penetrate” between enterocytes, through the basement membrane, and are able to form holes in it (reviewed by [6]). As a result of the contraction of the actomyosin cuff located around the IDCs, mChMs are squeezed out to the BM and then penetrate into the interstitial space through the pores in it.

As in adult rats, in newborns, there are two populations of ECs throughout the lymphatic vessels, one that has a basement membrane and the other that has sinusoidal contacts and an intermittent BM [3,30]. The crucial role of intestinal lymphatic vessels has been illustrated by studies of mice expressing the diphtheria toxin receptor under control of the lymphatic vessel endothelial hyaluronic acid receptor 1 (LYVE1) promoter. Although lymphatic capillaries of all tissues express LYVE1, diphtheria toxin administration selectively ablated only intestinal and lymph node lymphatics within 24 h. Loss of intestinal lymphatics resulted in severe gut inflammation, sepsis and high lethality [37]. In mice, duodenal and ileal villi have on average two and one LCs, respectively [23]. 

During the first feeding, ChMs can enter the blood capillaries, although rarely. In newborns, there is penetration of chylomicrons into blood capillaries in which there is not enough fenestrae. A comparison of the phenotypes of enterocytes studied during the first feeding shows that they more clearly correspond to the phenotype that is obtained when adult enterocytes are overloaded with lipids. The first feeding of the newly born also leads to the accumulation of lipid droplets in the cytoplasm of enterocytes, overload of the Golgi complex and the appearance of draining lipid droplets, as well as giant chylomicrons in the space between enterocytes and in the interstitial. 

In the intercellular spaces, the ChMs merge to form giant ChMs. These giant ChMs are hardly pushed into the interstitial space and are slowly absorbed by the LCs through their valve-like inter-endothelial contacts. In newborns, clusters of large mChMs were observed in the interstitial plate of the intestinal villi. mChMs within the diameters of 150–200 nm were detected in the immediate vicinity of LCs, as well as in the contact zone of endothelial cells. In contrast, in adult rats, under conditions of high lipid load, large, fused lipid particles are not found in the interstitial space. 

Additionally, when animals were overloaded with lipids, large lipid droplets are formed in the interstitial space, which are poorly absorbed in lipid capillaries. Why do mChMs fuse? We propose that this could be a result of errors in the glycosylation of apoproteins in the post-Golgi compartments, because lipid overload is also associated with the slowing down of intra-Golgi transport and delay of the mChM exit from the post-Golgi compartments containing LAMP2 and Neu1. Acidic environments and presence of neuraminidases could induce desialylation of mChMs, which is important for atherogenesis. As a result, the negative charge on the surface of mChMs decreases [5,6,37]. 

Sialylation plays an essential role in protecting mucus barrier integrity from bacterial degradation and is governed by ST6GALNAC1 (ST6), a local sialyltransferase in the gut [59]. mChMs were not labeled with lectin specific for sialic acid (our unpublished observations, the paper is submitted). A phenotype similar to the one described by us when feeding newborns and overloading the enterocytes of adult animals with lipids was found in mice with a deficiency of a transcription factor similar to the pleomorphic adenoma gene 2 [60].

Of interest, in response to an acute dietary fat challenge, Dgatl^−/−^ mice have abnormal accumulation of lipid droplets within enterocytes and a slower rate of TAG secretion into the blood [27,47]. Mice with deficiency in acyl CoA: diacylglycerol acyltransferase 1 (Dgat1^−/−^ mice) have a reduced rate of intestinal TAG secretion and abnormal TAG accumulation in enterocyte dCLDs [61].

ChMs are absorbed by LCs of the intestinal villus, which suck them due to the operation of the containing intramural valves within the contacts between ECs of LCs [4,30]. Sometimes, mChMs directly entered the blood capillaries already in the villi. However, the ability of the lymph node macrophages to appear within lymph node sinuses and consume large mChMs represents the specific filter preventing the appearance of large mChMs in the blood.

In LCs, the structure of inter-endothelial contacts was similar to those in adult animals. A unique feature of lacteals is their LCs are closely associated with villus SMCs [22,23]. Villi undergo constant movement owing to intestinal wall peristalsis and villus longitudinal and/or lateral contractions [22,23,24,25]. There is a work showing that in mice, the increase in the development of tight junctions between endothelial cells of lymphatic capillaries in the intestinal villus leads to the resistance of mice to an excess of food [62].

The role of the affinity of receptors to Fc fragments of IgG is important for apical endocytosis. Indeed, IgG-Fc-coated particulates, including even erythrocytes, are subjected to persorption from intestinal lumen via villous columnar epithelial cells in the small intestine [63]. In early embryonic development, access of maternally deposited nuclear proteins to the genome is temporally ordered via importin affinities [64]. One pathway relies on lipid exchange at sites of close contact between different membrane-bound organelles, sites now established as specific membrane contact sites. Sites of ER–endolysosomal contact serve as “metabolic platforms” for regulating inter-organelle lipid trafficking [65]. On the other hand, the flippase protein is among the essential for the life of a minimal cell [66]. Therefore, we believe that flip-flop plays an important role in the transcytosis of lipids across enterocytes. Recently, it was established that rat heart muscle cells (cardiomyocytes) undergo a 63% decrease in nuclear pore numbers during maturation, and this changes their responses to extracellular signals. The maturation-associated decline in nuclear pore numbers is associated with lower nuclear import of signaling proteins such as mitogen-activated protein kinase. Experimental reduction in nuclear pore numbers decreased nuclear import of signaling proteins, resulting in decreased expression of immediate-early genes [67]. Similarly, reduction in the nuclear pores in enterocytes could be responsible for the inhibition of apical endocytosis in adult rats.

Summaries of our observations are presented in Table 4 and Figure 9 and Figure 10. However, additional experiments are required. Indeed, after this work, many questions arise that require further study:Identification of proteins that transfer lipids from the BLPM to the smooth ER membrane and from the endosome membrane, as in newborns, into the membrane of smooth ER attached to endosomes.Which proteins in enterocytes are responsible for the attachment of smooth ER to the BLPM or to the membrane of endosomes?Why is the division of enterocytes blocked on the intestinal villi itself in adult animals and remains only in crypts?Is there a lot of cholesterol in mother’s milk immediately after birth?Which fatty acids prevail in breast milk: short or long carbon chain?Where does cholesterol come from for the synthesis of chylomicrons in newborns?How much cholesterol do the livers of newborns synthesize?Why do newborns not have a smooth ER attached to the BLPM?We have shown that endocytosis in non-fed baby rats leads to the capture of membranes from the intestinal lumen; these membranes enter the lumen of the Golgi apparatus and then are secreted into the space between enterocytes and, apparently, are captured in the lymph and then they get into the blood. Consequently, for some time, membranes from mother’s milk or even proteins can be endocytosed and, thereby, the newborn will form an immunological tolerance to the mother’s proteins. This hypothesis should be tested by immunologists.It could be that as maternal breast milk is stopped to be deliver, fewer Fc fragments of IgG antibodies enter the lumen of the small intestine. This leads to the fact that the binding of ligands to the receptors of these fragments decreases sharply, signaling decreases and the synthesis of receptors decreases. Therefore, gradually, the severity of apical endocytosis decreases. There is a hypothesis that it is all about the receipt of IgG-class antibodies. This can be checked if one gave these antibodies to young animals for longer, or if one fed them immediately with a glucose solution, for example, and starch without adding antibodies and maybe with the addition of a minimum amount of lipids. What happens if adult mice are fed fragments of IgG? Will sufficiently developed apical endocytosis not reappear?

Preliminary recommendations that require careful verification include the following: 1. do not feed whole mother’s milk immediately; it needs to be diluted and given little by little. 2. Reduce the amount of lipids in mixtures; this is especially important for the initial period of life. New directions of biological science are formed.

## 4. Material and Methods

Methodology and details of all ethics rules were described in [3]. Briefly: All those experimental animal procedures were approved by the Committees of the Ivanovo State Medical Academy. Wistar rats were obtained from the Moscow Cardiological Center (they took them from Taconic Farms (Germantown, NY, USA)) and were maintained either on Purina rodent chow (no. 5001 ICN Pharmaceuticals, Inc., Cleveland, OH, USA) or using the manually prepared food corresponding to the standards. All procedures were in accordance with EU directive 2010/63/EU. The animal facilities of the St. Petersburg Pediatric University and Ivanovo State Medical Academy housed animals in plastic sawdust-covered cages on a 12 h/dark/light cycle, keeping them under standard conditions (at room temperature and fed standard rat pelleted food and water ad libitum). 

Six Wistar newborn rats were taken immediately after their birth and six Wistar newborn rats were examined when its stomach appeared full after feeding. Six Wistar 6-month-old male rats were used for the examination of lipid overloading and six Wistar 6-month-old male rats were used as their control. Rats were anesthetized with a combination of Zoletil (the active substances: zolazepam hydrochloride, thiamine hydrochloride in equal proportions) and 2% Rometar (the active ingredient is xylazine hydrochloride) in the ratio of 3:1, in a dose of 0.1 mL per 100 g of body weight [3].

Animals were removed from the experiment before the end of anesthesia after opening the chest by the intracranial administration of a saturated solution of potassium chloride at a dose of 1–2 mM/kg [41]. Trained persons sacrificed rats. Death was confirmed observing cessation of heartbeat and respiration, and absence of reflexes, in agreement with international standards (https://www.lal.org.uk. access date: the 20 March 2022) While the animals were under ether anesthesia, jejune tissue was removed, processed, embedded, sectioned and stained. All experimental animal procedures were approved by the Committees of the Ivanovo State Medical Academy and St. Petersburg State Pediatric University. The procedures for animal use were conducted in accordance with the ethical and legal standards of the Russian Federation mentioned in Order no. 755 of the Ministry of Health of the USSR of 12 August 1977. “On measures to further improve the organizational forms of work using experimental animals” and a letter from the Ministry of Agriculture dated 5 February 2022 no. 13-03-2/358, “On modern alternatives to the use of animals in the educational process” and 2010/63/EU legislation on animal protection. The experiments were approved by the decision of the Academic Council of St. Petersburg Pediatric University no. 10 from 23 September 2015 and decision of ethic committee of Ivanovo State Medical Academy (no. 1 from 5/XII, 2018) in compliance with the above-mentioned Order no. 755 of the Ministry of Health of the USSR of 12 August 1977. “On measures to further improve the organizational forms of work using experimental animals” and a letter from the Ministry of Agriculture dated 5 February 2022 no. 13-03-2/358, “On modern alternatives to the use of animals in the educational process”. All experiments on live animals were carried out in Russia; samples irreversibly fixed with glutaraldehyde, embedded in Epon or gelatin (with subsequent fixation) in Russia and only then were plastic samples transported to Italy, where these plastic samples were examined.

Trained persons sacrificed rats. Death was confirmed observing cessation of heartbeat and respiration, and absence of reflexes, in agreement with international standards (https://www.lal.org.uk; access date: the 20 March 2022). While the animals were under ether anesthesia, jejune tissue was removed, processed, embedded, sectioned and stained. After injection of anesthetic, the abdomen of the animal was opened and the initial part of the jejune was cut and placed into fixative. All those experimental animal procedures were approved by the Committees of the Ivanovo State Medical Academy (see ethic statement in [3]). Overloading of adult rats with oil and sample preparations were performed exactly as it was described by [3,42,43,44,45]. Lymph nodes observed just below the initial segment of jejunum were cut and fixed. Methods of analysis area described previously [46]. 

Rabbit polyclonal antibody against CENPF (Anti-CENPF antibody [ab5]) was obtained from Abcam (Cambridge, UK; catalog no. ab84697). Polyclonal Anti-PCNA antibody was from Sigma-Aldrich (Milan, Italy; catalog no. AV03018-100UG). Anti-cyclin A polyclonal antibody was from ThermoFisher Scientific (Newark, DE, USA; catalog no. PA5-16516). The Invitrogen NEU1 Polyclonal Antibody was from ThermoFisher Scientific (Newark, DE, USA; catalog no. PAS-42552). Invitrogen rabbit polyclonal Antibody against Lysosomal-associated membrane protein 2 (LAMP2) was from ThermoFisher (Newark, DE, USA; catalog no. PA1-655). Lectin Limax flavus (LFA) conjugated with biotin was from MY BioSource (in Italy: ITALIA Gentaur SRL, Bergamo; catalog no. MBS656449). Streptavidin Gold Conjugates (20 nm, 10 OD) and 10 nm, 10 OD) were from Abcam (Cambridge, UK; catalog no. ab270029 and ab270041, correspondingly). Rabbit polyclonal antibody against NEU1 was from ThermoFisher Scientific (Newark, DE, USA; catalog no. PA5-42552). Gum arabic was from Sigma–Aldrich (Milan, Italy; catalog no. G9752). Thiocarbohydrazide was from Sigma-Aldrich (Milan, Italy; catalog no. 223220). The 3 mm diameter copper meshed grids with a holder (Special pattern with handles 100 mesh coordinate Copper) were from Labtech Serving scientists (Heathfield, East Sussex, UK; catalog no. [KU]: 07D00937). Veco Square Mesh Handle 150 mesh Copper Grids were from Fisherscientific (Waltham, MA, USA; catalog no. 50-289-1).

A Zeiss LSM510 laser scanning confocal microscope was used for examination of samples containing the monolayer of the aortic ECs. Samples were examined by scanning and transmission electron microscopy exactly as was described [35,38,50]. Briefly: the aorta was cut with a fresh razor blade at the level of the middle of the distance between the center of the frozen zone and its edge. Then, with cactus needles, the vessel was pinned to a thin plate of bark and subjected to drying by passing through the critical point. The posterior wall of the aorta was cut lengthwise and the sample was attached with cactus needles to a thin parallelepiped cut from the cork bark, and then dehydrated and dried by passing through the critical point. A thin layer of gold was sprayed onto the surface of the endothelium. At the same time, in places where the sample had undercut edges, conductive bridges were formed with the help of silver glue, which allowed electric charges to drain onto the sample holder.

After fixation of samples with 2.5% glutaraldehyde in 0.1 M cacodylate buffer (pH 7.4) they were post-fixed. Initially samples were washed with 0.15 M sodium cacodylate buffer followed by incubation in the reduced OsO_4_ for 1 h on ice. After washing in distilled water, the samples were incubated again in 0.3% thiocarbohydrazide for 20 min, washed with distilled water, and finally incubated a third time in 2% OsO_4_ in water for 30 min [4,19,20,21,22]. Immune labeling of cryosection and their acquisition was made exactly as described [46,54]. Fixed samples of lymphatic nodes were dehydrated and dried transferring samples through the critical point, evaporated with gold and examined in the scanning electron microscope Hitachi-S-570 (Hitachi, Tokyo, Japan) as described by [68,69].

For the pre-embedding-based immune electron microscopy, samples were washed from fixative in buffer, incubated with the blocking buffered solution for 1 h, rinsed four times for 10 min, incubated with primary antibody dissolved in blocking solution for 4 h at room temperature. Then four blocking buffer rinses over 30 min were performed and the secondary antibody incubation was applied: species-specific Fab fragments antibody labeled with 1.4 nm nanogold in blocking solution overnight at room temperature and second fixation: 2% glutaraldehyde in 0.1 M sodium cacodylate buffer (pH 7.4) for 15 min occurred. Further samples were washed in the HEPES buffer: 50 mM HEPES with 200 mM sucrose, pH 5.8, four times over 30 min. Then, incubation in the complete silver enhancer solution for 3–20 min (shielded from light or under dark-red light) and the neutral fixer solution composed of 250 mM sodium thiosulfate and 20 mM HEPES at pH 7.4 were applied. To stop the enhancement: three rinses were used over 15 min until gum arabic was gone. Next, samples were treated with 0.1% OsO_4_ for 30 min, subjected to dehydration and embedded into Epon [45]. 

The microvascular corrosion casting/SEM method was performed as described [70,71,72] Briefly: The animals were anesthetized as it was described above. The casting media was injected into the heart using a syringe. The injection pressure was equal to 80 mm Hg. The complete removal of the blood was performed. The circulatory system was rinsed with 37 °C heparinized Tyrode^®^ solution (5000 IU/l) until the efflux of the incised portions of several subcutaneous venae of animal limbs was clear. Immediately after the pre-casting treatment, the injection medium was prepared by adding to the main reagent (resin) a catalyst/accelerator to initiate polymerization. Thereafter, we injected Mercox II Red (from LARD Research Industries; catalog number 21,245–Mercox II Red–Kit). Animals’ bodies were left two hours at room temperature, then put in a 60 °C water bath for 24 h to accelerate and complete the polymerization of the perfused casting medium. Small intestine was removed and macerated in 15% KOH solution at 40 °C for 2 days. The specimens were cleaned in 2% formic acid. In order to remove the surrounding tissues, the injected specimen was immersed into the 15–20% sodium hydroxide or potassium hydroxide solution (at 60 °C, for 24 h). In order to remove the white saponified materials resulting from the maceration of tissues rich in lipids with sodium hydroxide, casts were washed in running water. Then, microdissection was carried out to expose the structures of interest. The vascular casts were dried in air without any detectable distortion or dislocation of the fragile parts of the casts. In some cases, we used freeze-drying to minimize surface tension. The specimen was coated with a heavy metal (gold), and then observed in SEM with an accelerating voltage of 5–10 kV. Then, SEM observation of corrosion casts was performed.

Pre-IEM based on gold-enhancement was performed according to He et al. [73] with small modifications. Briefly, after fixation of cells with glutaraldehyde (see above), samples were washed with the blocking buffered solution (four rinses over 30 min), incubated with primary antibody dissolved in blocking solution for 4 h at room temperature, rinsed with blocking buffer (four times over 30 min), incubated with the species-specific Fab fragments of secondary antibody labeled with 1.4 nm nanogold in blocking solution overnight at room temperature. Then, cells were fixed additionally with 1.6% glutaraldehyde in 0.1 M sodium cacodylate buffer (pH 7.4) for 15 min, rinsed with HEPES buffer (50 mM HEPES with 200 mM sucrose, pH5.8, four times over 30 min), washed 3 × 5 min with PBS including glycine (20 mM sodium phosphate, pH 7.4, 150 mM NaCl, 50 mM glycine) to remove aldehydes, rinsed (3 × 5 min) with PBS–BSA–Tween (PBS containing 1% BSA and 0.05% Tween 20), and washed (3 × 5 min) with Solution E (5 mM sodium phosphate, pH 5.5, 100 mM NaCl) from the gold enhancement kit (GoldEnhance-EM 2113; Nanoprobes, Inc., Yaphank, NY, USA). Next, samples were placed in a mixture of the manufacturer’s Solutions A and B at a 2:1 ratio (80 µL of A and 40 µL of B for 5 min and 200 µL of Solution E with 20% gum arabic (Sigma–Aldrich) and then 80 µL of Solution C were added in order to develop gold for 7–15 min. The enhancement was conducted at 4 °C. Further, samples were transferred to the neutral fixer solution composed of 250 mM sodium thiosulfate and 20 mM HEPES at pH 7.4 to stop the enhancement (three rinses over 5 min), washed with buffer E for 3–5 min, incubated in 1% OsO4 in 0.1 M sodium phosphate (pH 6.1) for 60 min and rinsed with distilled water. Finally, after standard dehydrations, cells were embedded into Epon. Semi-thin sections were cut, achieving an exit to a depth of no more than 5 microns because only at this depth do antibody solutions penetrate in cells. Immunolabeling was determined at a depth of up to 5 microns from the cut surface. After finding a cell with immunolabeling, serial ultrathin sections were made, and the cell phase was determined based on the number of centrioles.

We compared newborn rats and the newborn rats after the first feeding. Six pairs of animals composed of control and experimental rats were formed. Then, the average percentage of positive samples was calculated for the experimental and control animals. These mean values were considered as the statistical units.

In order to estimate the percentage of cells with defined phenotype, we used blind analysis and examined 3–4 cells in each member of the pair. To test whether differences were significant (*p* < 0.05), Student’s t-tests, paired t-tests and non-parametric Mann–Whitney U tests were used. Normality of the datasets was assessed using Shapiro–Wilk normality tests. In majority of cases, we used nonparametric Mann–Whitney U test. Values are mean ± SD of 6 variants (*n* = 6). In the majority of cases, we used non-parametric Mann–Whitney U tests. A difference was considered significant when *p* < 0.05. The standard software package GraphPad Prism (Prism: Version 9.4.2) was used. Data are given as means ±standard deviation (SD). In the text, the words “differ”, “smaller” or “higher” indicate that two values are significantly (*p* < 0.05) different [47,48,74].

## Figures and Tables

**Figure 1 ijms-23-14179-f001:**
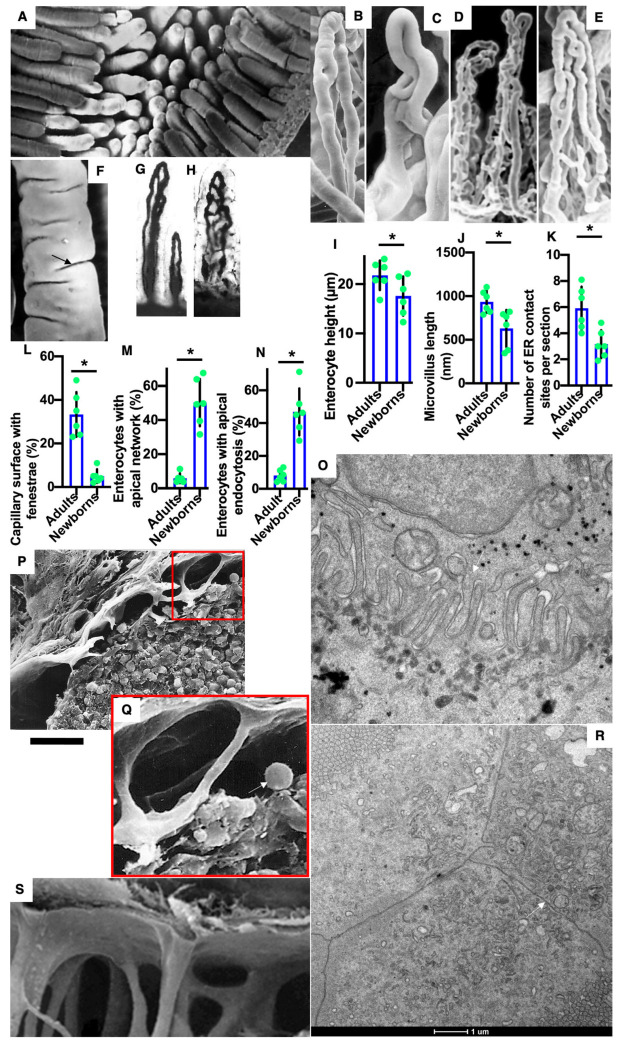
Structure of enterocytes, vascular beds and lymph nodes in newborn rats before their feeding. (**A**,**F**) Scanning electron microscopy of the luminal surface of intestinal villi. (**B**–**H**) The structure of the vascular bed of the intestinal villi of newborn rats before their feeding. (**B**–**E**) Corrosion casts of blood microvessels in intestinal villus of newborn rats. (**G**,**H**) Total preparations of intestinal villi after staining for horseradish peroxidase. Blood capillaries contain black precipitate of DAB-HRP reaction in their lumen. (**I**–**N**) Graphs describing quantitative data of enterocytes. The percentage of enterocytes with a pronounced subapical network is higher in newborn rats. (**O**) Large interdigital contact (arrow) between two enterocytes in newborn rat before its feeding. (**P**,**Q**,**S**) Scanning EM of the subcapsular sinus of lymph node shows that before the first feeding of newborn rats, this sinus is almost empty and contained lower number of macrophages (arrow). This marker (gold dots) is visible at the trans side of Golgi stacks. (**R**) Simple contacts (arrow) between enterocytes without IDCs. *with the horizontal line below indicates that these two bars are different (*p*<0.05). Scale bars: 200 µm (**A**); 80 µm (**B**–**E**); 25 µm (**F**); 103 µm (**G**,**H**); 15 µm (**P**); 270 nm (**O**); 5 µm (**Q**). In (**R**), the scale bar is equal to 1 µm (indicated below it).

**Figure 2 ijms-23-14179-f002:**
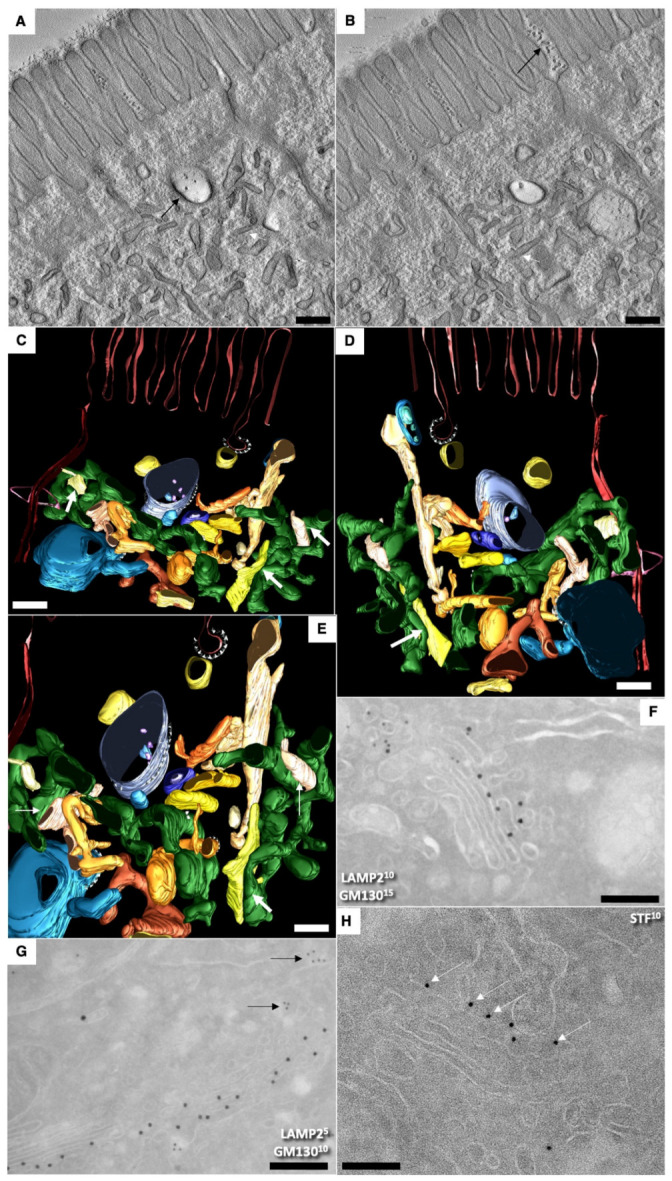
Structure of enterocytes in newborn rats before their first feeding. (**A**–**F**) Subapical membranous tubular network (white arrows). Black arrows indicate membrane remnants between microvilli and inside endosome (**A**). (**A**,**B**) Serial tomographic images. (**C**–**F**) A three-dimensional model obtained on the basis of EM tomography. View from different angles (**F**). The enlarged area of the image enclosed inside the red box in (**D**). ER is colored green. Rough ER is large vacuoles and cisterns. Smooth ER is tubes. Endosomal structures are yellow. The purple color indicates APM. The BLPM is indicated in red. Multivesicular corpuscles and late endosomes are indicated in blue. Endosomal–ER contact sites are shown with white arrows. Clathrin—purple triangles. The Golgi stacks were already polarized. The immune EM marking for GM130 is present on the side of the Golgi complex (see movie 1). (**G**,**F**) Labeling for LAMP2 and GM130 (**G**) in (**G**) and LAMP2 (10 nm ) and Golgin-97 (15 nm) on (**F**). (**H**) Immune EM labeling for sialyltransferase (STF) of the enterocytes Golgi stack. Scale bars: 270 nm (**G**); 210 nm (**F**); 230 nm (**H**). In (**A**–**E**), scale bars are equal to 250 nm. In (**E**), the scale bar is equal to 70 nm.

**Figure 3 ijms-23-14179-f003:**
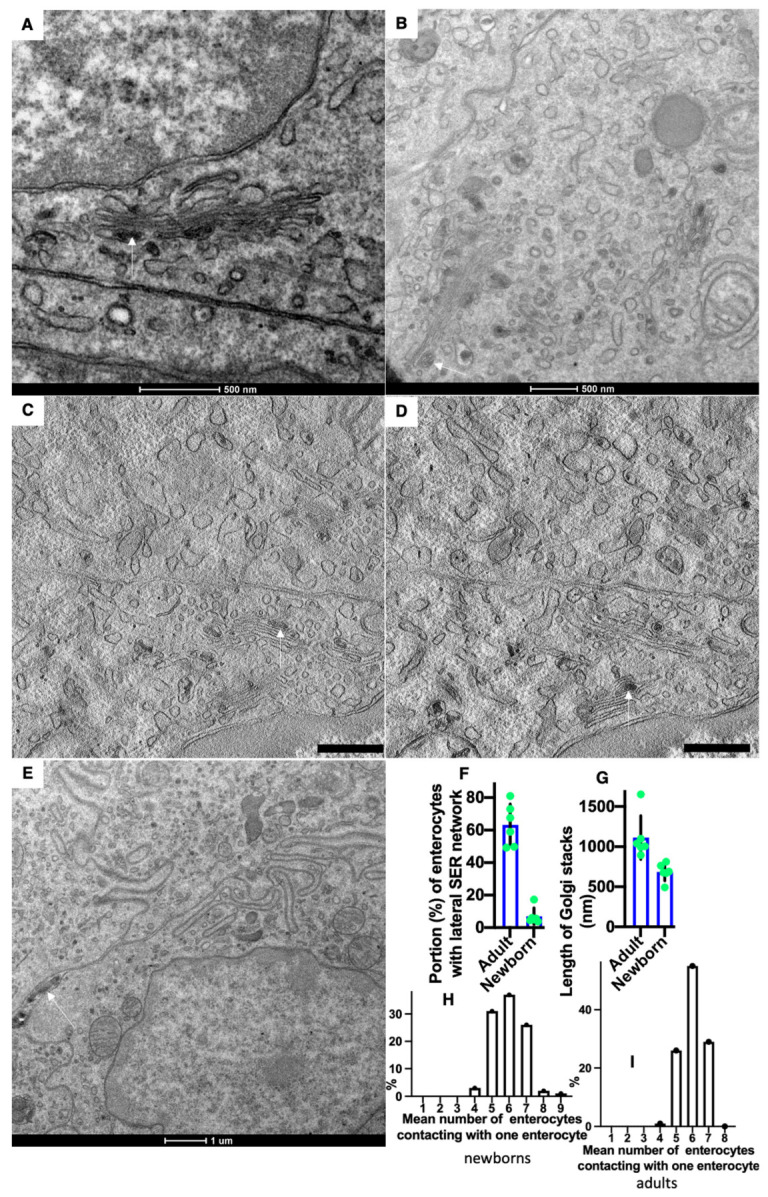
Structure of Golgi complex (GC) and vascular walls in newborn rats before their first feeding. (**A**,**B**) Routine Epon sections. (**C**,**D**) Serial tomographic slices. Multilamellar structures (arrows) are visible in the cisternal distension of the GCs. (**E**) Membrane remnants (white arrow) are visible inside the contact space between enterocytes. (**F**) The graph shows that in newborn rats, lower (*p* < 0.05) number of enterocytes exhibited lateral SER network. (**G**) The graph shows that in newborn rats, the length of Golgi stacks was lower (*p* < 0.05). (**H**,**I**) Graphs show the distribution of the mean number of neighbors of one enterocyte in newborn (**H**) and adult (**I**) rats. In the newborn rats, the shift of these distributions towards the low number is significant (*p* < 0.05). Scale bars: 465 nm (**C**,**D**). In (**A**,**B**,**E**), scale bars are indicated below each image.

**Figure 4 ijms-23-14179-f004:**
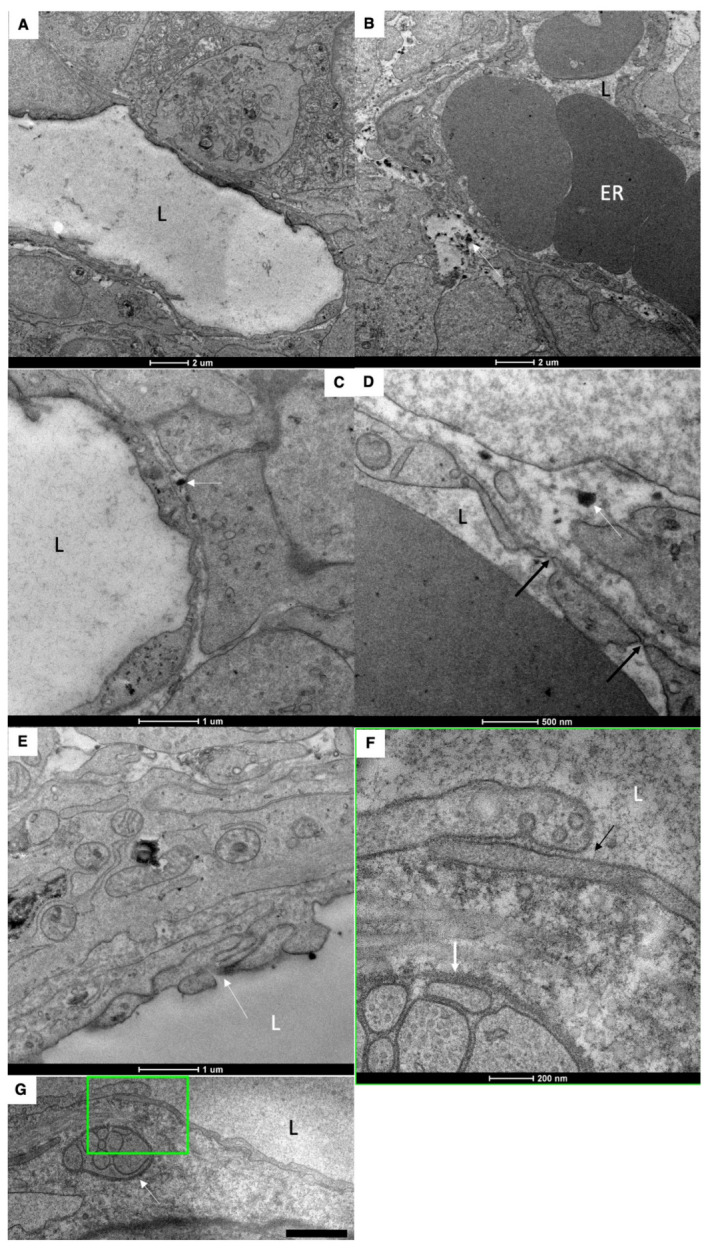
Structure of blood and lymph capillaries in newborn rats before their first feeding. (**A**–**C**) Blood capillary. Low number of fenestrae in capillary wall. (**D**) Low density of fenestrae (black arrows). White arrows in (**B**–**D**) show membrane remnants in the interstitial space. (**E**–**G**) EM images of the LC wall. White arrow in (**E**) shows complex inter-endothelial contact. White arrows in (**F**,**G**) indicate nervous terminal. The square box with green border it enlarged in (**F**). (**F**) Enlarged area inside the square box with green borders in (**G**). Black arrows indicate the simple tile-like contact between endothelial cells of LC. Low number of caveolae. L, lumen of blood and lymph capillaries. ER, erythrocytes. Scale bars: 990 nm (**G**). In (**A**–**F**), scale bars are indicated below images.

**Figure 5 ijms-23-14179-f005:**
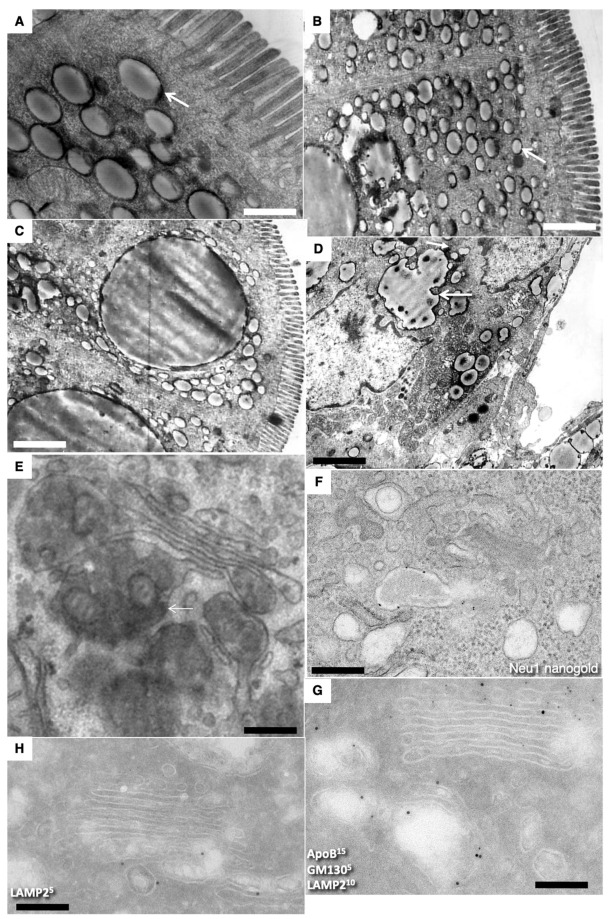
Structure of enterocytes in newborn rats after their first feeding. (**A**–**C**) Formation of large lipid droplets (white arrows) in the cytoplasm. (**D**) Formation of “lipid lakes” (black arrows) between enterocytes after the first feeding of a newborn rat. (**E**) Overloading of the GC. (**F**) Large ChMs in post-Golgi compartment positive for Nau1. (**G**,**H**) Labeling for LAMP2, GM130 and ApoB of post-Golgi carriers containing large ChMs. Scale bars: 510 nm (**A**); 1020 nm (**B**); 1.3 µm (**C**); 1025 nm (**D**); 240 nm (**E**); 340 nm (**F**).

**Figure 6 ijms-23-14179-f006:**
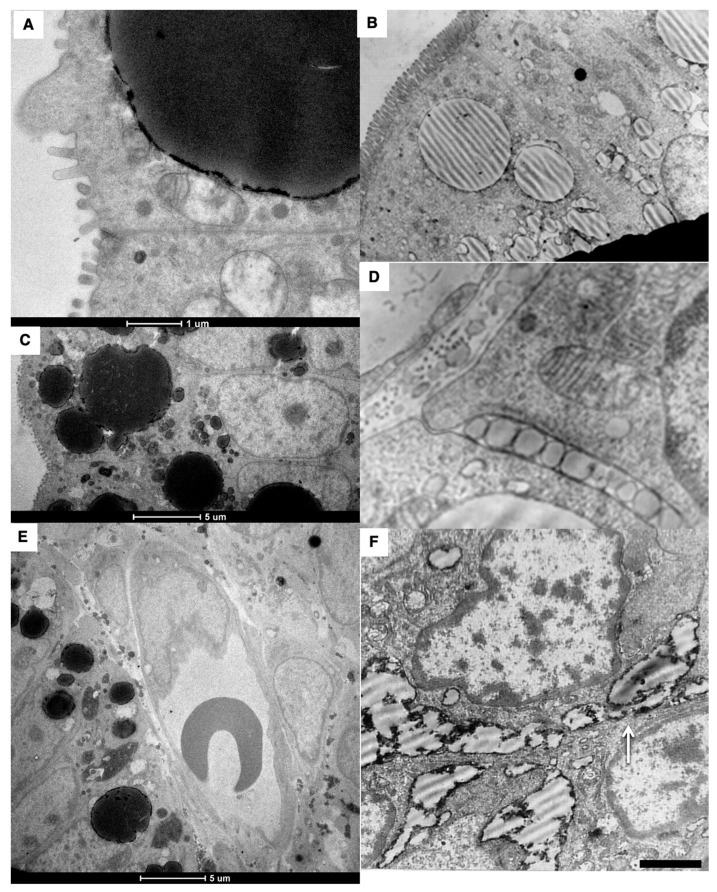
Structure of small intestine enterocytes in newborn rats after their first feeding. (**A**–**C**) Formation of huge droplets and “lipid lakes” in enterocytes of newborn rats after the first feeding. (**A**,**C**) Thick (200 nm) sections. (**D**) Large ChMs between enterocytes. (**E**) The lumen of the blood capillary with the erythrocytes inside its lumen does not contain chylomicrons. (**F**) “Lipid lakes” between enterocytes. Scale bars: 1050 nm (**B**); 290 nm (**D**); 610 nm (**F**). In (**A**,**C**,**E**), scale bars are indicated below each image.

**Figure 7 ijms-23-14179-f007:**
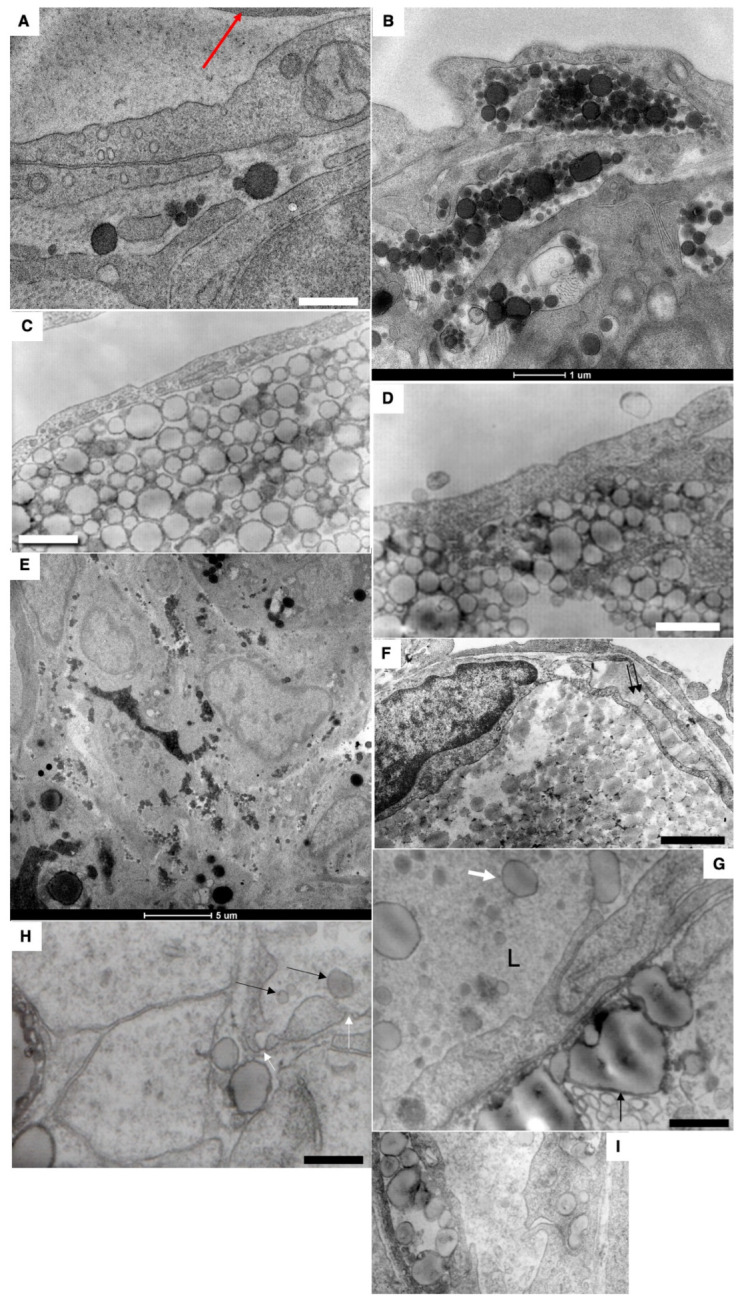
Passage (presumably) of large chylomicrons through intercellular spaces. (**A**) Large chylomicrons are not captured by blood capillaries (the red arrow shows erythrocyte). (**B**–**D**) Accumulation of chylomicrons in the interstitial space. (**E**). Accumulation of chylomicrons between enterocytes. (**F**,**G**) Uptake of chylomicrons into the lymphatic capillary. Double arrows in (**F**) show large chylomicron in the inter-endothelial contact. (**G**) Large ChMs in the lumen of the lymphatic capillary. (**H**) Rarely, ChMs (shown by black arrows) were seen in the blood capillary, in the wall of which single fenestrae are visible (shown by white arrows). (**I**) Chylomicrons (to the right) inside the vacuoles within cytoplasm of endothelial cells of lymphatic capillary. Scale bars: 595 nm (**A**,**H**); 610 nm (**C**): 460 nm (**D**,**I**); 900 nm (**F**); 570 nm (**G**). In (**B**,**E**), scale bars are indicated below images.

**Figure 8 ijms-23-14179-f008:**
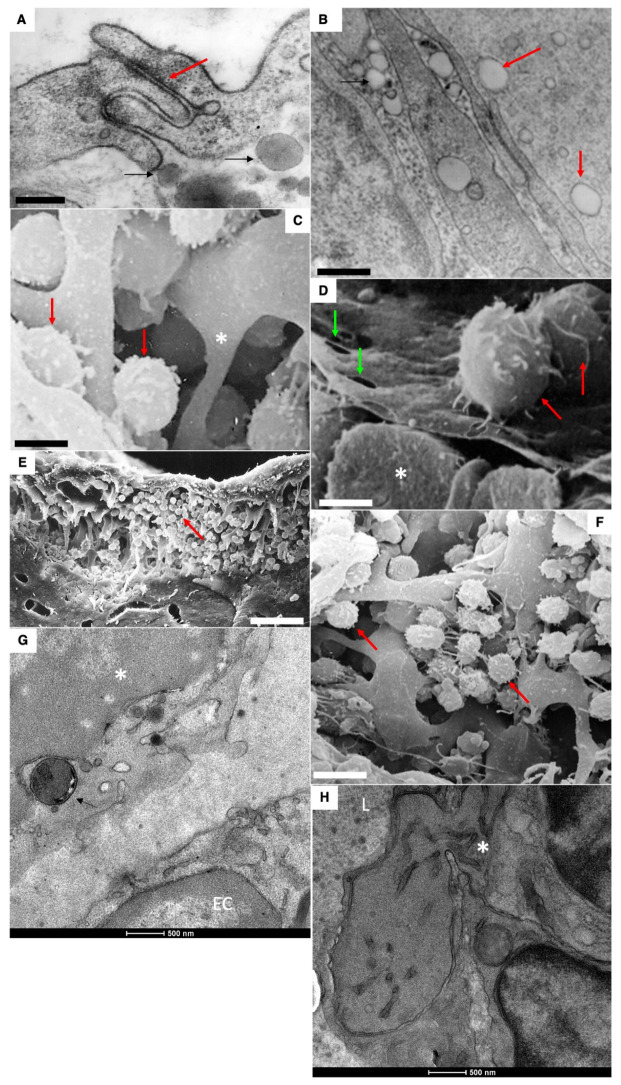
Mechanisms of resorption of large ChMs from the interstitial space into LCs of a newborn rat after feeding. (**A**,**B**) The lumen of LC is filled with ChMs (shown by the arrow). Large lipid droplets are visible in the interstitial (shown by the black arrows). (**B**) Interdigital contact between endothelial cells in LCs does not allow ChMs (arrows) to pass through it. (**C**–**H**) Migration of macrophages into subcapsular sinus of lymph node after the first feeding. Uptake of large chylomicrons by macrophages (**G**). Red arrows in all images indicate leucocytes. White asterisks demonstrate monocytes. Green arrows show pores in endothelial cells surrounding the sinus from the follicular side. Scale bars: 420 nm (**A**); 560 nm (**B**); 5.4 µm (**C**); 4 µm (**D**); 30 µm (**E**); 10.8 µm (**F**). In images (**G**,**H**), scale bars are below images and are equal to 500 nm.

**Figure 9 ijms-23-14179-f009:**
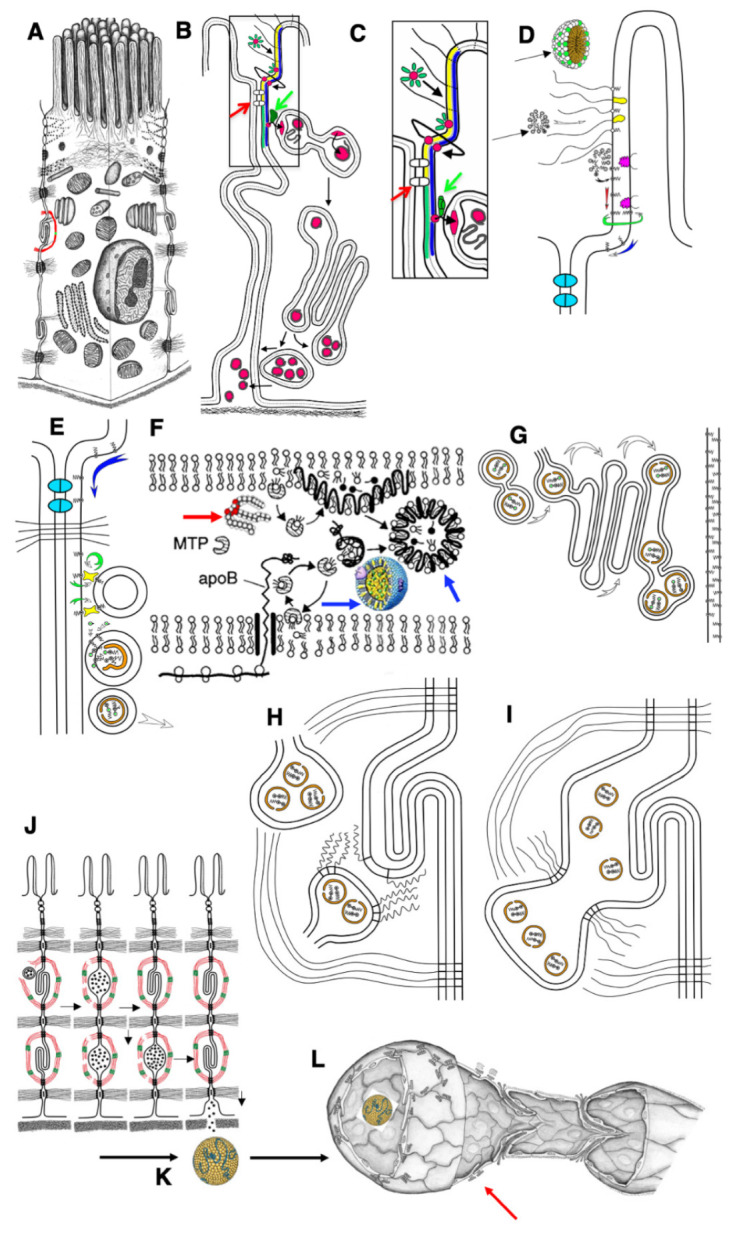
Schemes of the mechanisms of lipid transcytosis across enterocytes and their uptake by lymphatic capillary in adult rats (adapted from [4]). (**A**) Enterocyte. (**B**) The gut micelle (red ring surrounded with green ovals) is composed of free fatty acids (FFAs), cholesterol and bile acids (green ovals). The micelle passes through glycocalyx (black lines) and contacts with the external leaflet of the apical plasma membrane (APM; yellow layer). Then, FFAs and cholesterol are subjected to flip-flop (the bent line shows this path of red ring) and appear in the cytosolic leaflet of the APM (blue layer). (**C,D**) Higher magnification of the pathways shown in (**B**). (**D**) The micelles (black arrows) pass through glycocalyx (white arrow). Membrane proteins containing long polysaccharide chains of glycocalyx are yellow. Caveolin is magenta. FFAs (dots, black center) are inserted into the lipid bilayer of the APM, and then diffuse along the external leaflet (red arrow) or flip-flop (green arrow) and then diffuse along the cytosolic leaflet (blue arrow). This pathway allows FFAs to reach the basolateral plasma membrane because the FFA diffusion along the external leaflet is restricted by proteins of tight junctions (indicated with red arrows in (**B**) and (**C**) or colored in light blue in (**D**). (**C**) FFAs bypass tight junctions (white dots situated between APM are indicated with red arrow) and reach the sites where cisternae of the smooth endoplasmic reticulum are attached to the basolateral PM along with lipid transfer proteins (green dot indicated with green arrows). (**E**) These proteins (green arrows in (**B**) and (**C**)) constantly (circular green arrow) transfer FFAs and cholesterol through cytosol into the cytosolic leaflet of the smooth endoplasmic reticulum membrane (the arc-like green arrow). (**F**) Then, FFAs are transformed into triacylglycerols and cholesterol ethers (double-headed structures, one green dot in (**E**)). Finally, these are extracted from smooth endoplasmic reticulum membranes and ApoB (orange) forms pre-chylomicrons. (**D**) Diffusion of FFAs along PM. (**E**) Delivery of lipids into membrane of smooth ER attached to the BLPM of enterocyte. (**F**) ApoB protein (black line) is synthesized by ribosome (white double rings below lipid bilayer of the ER). Then triacylglycerols (thick red arrow) and cholesterol ethers are captured by ApoB and MTP protein and the chylomicrons (blue arrows) are formed. (**G**) Pre-chylomicrons are transported from the ER at the Golgi complex, where mature chylomicrons are formed and concentrated in distensions and at the trans side (white arrows). (**H**) Post-Golgi carriers are formed, connected with the Golgi complex, and then fuse with the basolateral plasma membrane of the interdigitating contacts with the help of SNAREs (zip-like lines). (**I**) Chylomicrons are in the extracellular space within interdigitating contacts. (**J**) Contraction of actin–myosin cuff (red–green) induces movement of chylomicrons towards basolateral membrane and their passage through the BM pores. (**K**) Chylomicrons (yellow) are delivered to the interstitial space. (**L**) Chylomicrons are captured by lymphatic capillary (red arrow) and delivered into its lumen through the inter-endothelial contacts of endothelial cells.

**Figure 10 ijms-23-14179-f010:**
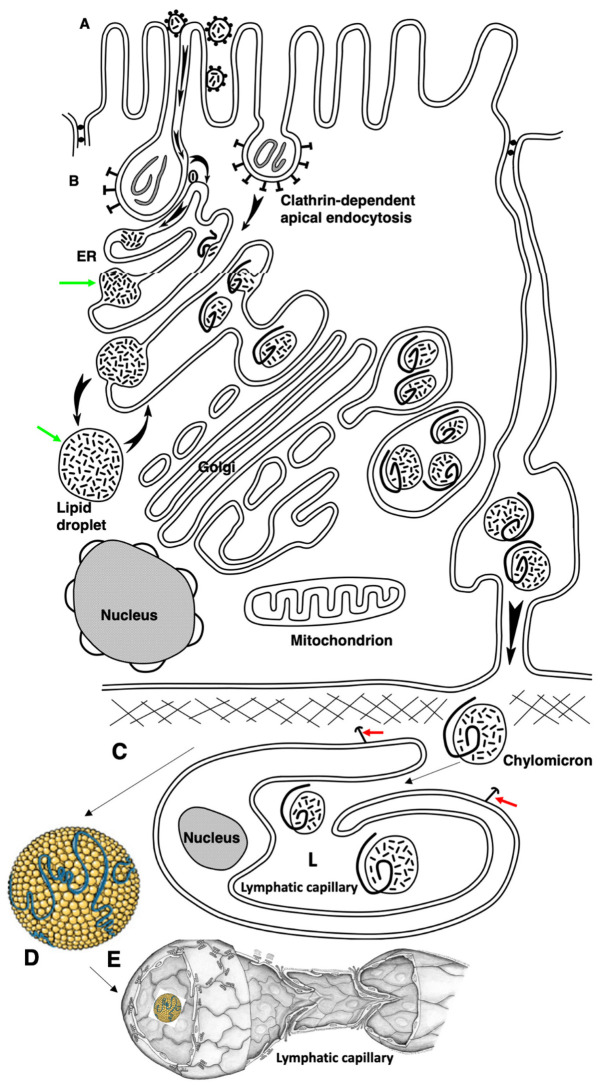
The hypothetical scheme of the lipid transcytosis in enterocyte of the newborn rats. (**A**) Micelles containing lipids and surrounded with bile acid molecules (black dots) passed through glycocalyx and contact with the apical plasma membrane of microvilli. Then FFAs and cholesterol are subjected to flip-flop. During apical endocytosis, the membrane remnants (grey lines inside endosomes) could be captured from gut. Next, FFAs and cholesterol diffuse along the cytosolic leaflet to the contact sites between membranes of endosome and the ER. (**B**) In endosomes, FFAs and cholesterol are captured by the yet-unknown lipid transfer protein(s) (oval dots with a line inside) and appear in the membrane of the ER. Green arrows show the ER-derived lipid droplets. Then, their pathway is identical to that in adult rats (see Figure 10). The chylomicrons are formed with the help of ApoB. Alternatively, triacylglycerols and cholesterol ethers are accumulated inside the membrane of the ER and the lipid droplets (black arrows) are formed. Chylomicrons in newborn rats are larger than those in adult rats when their enterocytes obtained normal amounts of lipid (see Figure 9). These chylomicrons passed across the Golgi complex and were secreted into interstitial space where they could fuse with each other. (**C**) Finally, large chylomicrons are captured by lymphatic capillaries and appear in their lumen (L). Red arrows indicate the anchor filaments. (**D**) Large chylomicron. (**E**) 3D model of lymphatic capillary.

**Table 1 ijms-23-14179-t001:** The ratio between the longitudinal and transverse diameters on the cross-sections of the intestinal villi of white rats (shape factor).

Villus Levels	Newborn Rats before Feeding	Newborn Rats after Feeding	Starved Adult Rats
Just below the enterocytes	1.7 ± 0.3 *	1.9 ± 0.2	2.8 ± 0.2 *
Middle	2 ± 0.2 **	1.9 ± 0.2	5.7 ± 0.3 **
Base	1.8 ± 0.3 ***	1.9 ± 0.2	12.6 ± 0.9 ***

*, ** and ***: difference between means is significant (*p* < 0.05).

**Table 2 ijms-23-14179-t002:** Number of blood capillaries on cross-sections of different levels of intestinal villi of white rats.

Villus Level	Newborn Rats before Feeding	Starved Adult Rats
Just below the enterocytes	2.00 ± 1.00	2.75 ± 0.25
Middle	5.80 ± 0.73 *	10.50 ± 0.50 *
Base	4.40 ± 0.92 **	8.25 ± 0.25 **

* and **: difference between means is significant (*p* < 0.05).

**Table 3 ijms-23-14179-t003:** Area of lymphatic capillary profiles on cross-sections of intestinal villi of white rats (in µm^2^).

Villus Level	Newborn Rats before Feeding	Newborn Rats after Feeding	Starved Adult Rats
Just below the enterocytes	-	-	97.7 ± 4.9
Middle	21.9 ± 2.5 *	28.2 ± 2.5	112.2 ± 6.3 *
Villus level	21.9 ± 1.4 **	24.7 ± 2.8	101.1 ± 6.2 **

* and **: difference between means is significant (*p* < 0.05).

**Table 4 ijms-23-14179-t004:** Differences between the components of the epithelial–lymphatic barrier of adult animals and newborns.

Features	Adult Rats	Newborn Rats
Apical endocytosis	Weakly developed	Well developed
Subapical network of the endosomes	Weakly developed	Well developed
The network of the cisterns of the smooth endoplasmic network in the supranuclear zone	Well developed	Weakly developed
The contractile cuff around the finger-shaped contacts of the basolateral PM	Well developed	Weakly developed
The height of the cells	22 µm	20 µm
Mean microvillus length	1020 nm	760 nm
Tissue mosaics	Hexagons	Shifted towards pentagons
The pores in the BM	Well developed	Weakly developed
The occurrence of dendritic cells in the epithelial layer	Often	Rarely
The development of fenestra fields in endotheliocytes of blood capillaries	Well developed	Not developed
The asymmetry of the arrangement of the fenestra fields	In the capillary wall areas looking at enterocytes, the number of fenestrae was much higher	Weakly developed
The development of intravenous valves in the lymphatic capillaries is poorly expressed	Well developed	Weakly developed

## Data Availability

Not applicable. This study did not generate new unique reagents.

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
