# Peer review of "Effect of the First Feeding on Enterocytes of Newborn Rats"

_ijms, 2022, doi:10.3390/ijms232214179_

Round 1

Reviewer 1 Report

Effects on the first feeding on enterocytes of newborn rats

Nikonova M, Irina S. Ssorova, Ivan dimov, Natalia Karelina and Alexander A. Mironov

The English used in this paper could be ameliorated. Some of the sentences are cryptic

In the abstract: 1) When lipids are too much

   2) The first feeding induced formation of the phenotype suggesting about the lipid overloading of the system

In order to help readers, the authors could include a figure of the proposed model revealing differences between newborn and adult rats.

Figure 1: misspelling Enterocytes with apicAl endocytosis, the A is missing

                 Enterocytes height, there is a mistake enterPcyte

Points are missing lane 15, 62, 153. Additional bracket lane 158 should be removed.

The provided images are of high quality and very informative. Overall, the results are descriptive. Although I agree with the conclusion, the take-home message of this paper has to be replaced in the context and the authors should better discuss the physiological relevance of their findings. As it is, it remains too descriptive.

Author Response

Reviewer 1

Effects on the first feeding on enterocytes of newborn rats Nikonova M., Irina S. Sesorova, Ivan Dimov, Natalia Karelina and Alexander A. Mironov.

  1. The English used in this paper could be ameliorated. Some of the sentences are cryptic. In the abstract: 1) When lipids are too much... 2) The first feeding induced formation of the phenotype suggesting about the lipid overloading of the system.

Reply: We rewrote these part of the abstract

  1. In order to help readers, the authors could include a figure of the proposed model revealing differences between newborn and adult rats.

Reply: We include this model into the figures.

  1. Figure 1: misspelling Enterocytes with apicAl endocytosis, the A is missing.

Reply: We corrected this.

  1. Enterocytes height, there is a mistake enterPcyte.

Reply: we corrected this mistake.

Points are missing lane 15, 62, 153. Additional bracket lane 158 should be removed.

Reply: we corrected these mistakes.

The provided images are of high quality and very informative. Overall, the results are descriptive. Although I agree with the conclusion, the take-home message of this paper has to be replaced in the context and the authors should better discuss the physiological relevance of their findings. As it is, it remains too descriptive.

Reply: We re-wrote the discussion part and results.

Reviewer 2 Report

The subject of the Manuscript entitled “Effect of the first feeding on enterocytes of newborn rats” by Maria Nikonova, Irina Sesorova, Ivan Dimov, Natalia Karelina and Alexander Mironov is relevant and can be considered for publication in MDPI IJMS after revision of the manuscript. In vivo studies represent a valuable source of information regarding occurring processes in a living system. Although, I suggest some points be revised or explained:

1.    The manuscript requires detailed technical editing in terms of typographical error correction, also, editing of the English language should be done. Also, the labeling of the figures should be corrected, for example in line 257 it is written “(shown by the white arrow)” however in Figure 8A arrows are black and there are no arrows in Figure 8B.

2.    Although authors performed experiments according to already published protocols it would be much easier for readers to follow and understand if authors could explain in brief performed work and used protocols and instrumentation?

Author Response

Reviewer 2

The subject of the Manuscript entitled “Effect of the first feeding on enterocytes of newborn rats” by Maria Nikonova, Irina Sesorova, Ivan Dimov, Natalia Karelina and Alexander Mironov is relevant and can be considered for publication in MDPI IJMS after revision of the manuscript. In vivo studies represent a valuable source of information regarding occurring processes in a living system. Although, I suggest some points be revised or explained:

  1. The manuscript requires detailed technical editing in terms of typographical error correction, also, editing of the English language should be done. Also, the labeling of the figures should be corrected, for example in line 257 it is written “(shown by the white arrow)” however in Figure 8A arrows are black and there are no arrows in Figure 8B.

Replay; We corrected these mistakes.

  1. Although authors performed experiments according to already published protocols it would be much easier for readers to follow and understand if authors could explain in brief performed work and used protocols and instrumentation?

Reply: We added these descriptions.

Reviewer 3 Report

The mucosa of the small intestine features thousands of fingerlike folds of tissue referred to as villi that are created during animal development.  These villi are designed to increase the tissue surface area available for nutrient absorption from the gut lumen. The principal cell type that make up villi are enterocytes. Enterocytes create an organized collection of specialized microvilli on their apical surface known as the intestinal brush border that allow them to mediate nutrient absorption.  Below the enterocytes is a developed system of blood and lymphatic capillaries.  This study focusses on the characterization of the gut-to-lymph barrier of rats after first feeding.

Although this study contains some beautiful microscopy, the narrative of the study is very hard to follow.  The results section seems to be almost a “list of observations” that do seem to be connected in a coherent fashion (at least, none that I could follow).  Many details are found out of place in the narrative (details normally found in the figure legends are found in the main narrative) which makes the study very confusing.  Furthermore, many sentences are incomplete sentences, that do not make any sense to the reader.   I feel that it would be well-worth the time for the authors to reorganize their manuscript, keeping in mind that they must provide a logical narrative in their results section for the reader.

Author Response

Reviewer 3

The mucosa of the small intestine features thousands of finger-like folds of tissue referred to as villi that are created during animal development.  These villi are designed to increase the tissue surface area available for nutrient absorption from the gut lumen. The principal cell type that make-up villi are enterocytes. Enterocytes create an organized collection of specialized microvilli on their apical surface known as the intestinal brush border that allow them to mediate nutrient absorption.  Below the enterocytes is a developed system of blood and lymphatic capillaries.  This study focusses on the characterization of the gut-to-lymph barrier of rats after first feeding.

Although this study contains some beautiful microscopy, the narrative of the study is very hard to follow.  The results section seems to be almost a “list of observations” that do seem to be connected in a coherent fashion (at least, none that I could follow).  Many details are found out of place in the narrative (details normally found in the figure legends are found in the main narrative) which makes the study very confusing.  Furthermore, many sentences are incomplete sentences, that do not make any sense to the reader.   I feel that it would be well-worth the time for the authors to reorganize their manuscript, keeping in mind that they must provide a logical narrative in their results section for the reader.

Replay: We rewrote Results and Discussion correcting these mistakes.

Round 2

Reviewer 3 Report

The edits have improved the manuscript.